

# Physics-inspired integrated space-time Artificial Neural Networks for regional groundwater flow modeling

Ali Ghaseminejad[1] and Venkatesh Uddameri[1]

[1]Department of Civil, Environmental and Construction Engineering, Texas Tech University, Lubbock, TX 79409-1023

**Correspondence:** Venkatesh Uddameri (venki.uddameri@ttu.edu)

**Abstract.** An integrated space-time Artificial Neural Network (ANN) model inspired by the governing groundwater flow equation was developed to test whether a single ANN is capable of modeling regional groundwater flow systems. Model-independent entropy measures and random forest (RF) based feature selection procedures were used to identify suitable inputs for ANNs. L2 regularization, 5-fold cross-validation, and adaptive stochastic gradient descent (ADAM) algorithm led to a parsimonious ANN model for a 30,691 $km^2$ agriculturally intensive area in the Ogallala Aquifer of Texas. The model testing at 38 independent wells during the 1956–2008 calibration period showed no overfitting issues and highlighted the model's ability to capture both the observed spatial dependence and temporal variability. The forecasting period (2009–2015) was marked by extreme climate variability in the region and served to evaluate the extrapolation capabilities of the model. While ANN models are universal interpolators, the model was able to capture the general trends and provided groundwater level estimates that were better than using historical means. Model sensitivity analysis indicated that pumping was the most sensitive process. Incorporation of spatial variability was more critical then capturing groundwater level persistence. The use of the standardized precipitation-evapotranspiration (SPEI) index as a surrogate for pumping was generally adequate but was unable to capture the heterogeneous groundwater extraction preferences of farmers under extreme climate conditions.

## 1 Introduction

There is a growing recognition that alternative, data-driven, model formulations can be employed to better capture nonlinear aquifer response dynamics (Obergfell et al., 2019; Roshni et al., 2019; Rinderer et al., 2018; Adamowski and Chan, 2011; Daliakopoulos et al., 2005). Artificial Neural Networks (ANNs) are a class of machine learning algorithms that are universal approximators capable of modeling highly nonlinear phenomena (Hornik et al., 1989). ANNs exhibit the ability to capture short-term volatility and long-term persistence effects often seen in groundwater time-series data either explicitly through input specification or implicitly using specific architectures (Principe et al., 2000). As such, ANNs have been widely used to model groundwater level changes at individual wells (Liu et al., 2018; Nizar Shamsuddin et al., 2017; Trichakis et al., 2011; Uddameri, 2007; Nayak et al., 2006). While forecasting water levels at individual wells is sufficient in certain groundwater applications, capturing the spatiotemporal dynamics of groundwater levels is critical for regional-scale aquifer management.

While regional groundwater flow models continue to be developed using physically based methods that integrate conservation principles and Darcy's law (Anderson et al., 2015; Harbaugh, 2005), attempts have also been made to capture spa-





tiotemporal groundwater behavior using ANNs (Sahoo et al., 2017a, b; Tapoglou et al., 2014; Nourani et al., 2011, 2008). Spatiotemporal modeling of groundwater levels using ANNs is often carried out using a two-stage process. In the first stage, separate ANN models are constructed at individual wells where groundwater level time-series data are available. In the second stage, forecasts from these individual wells are regionalized using statistical and geostatistical interpolation methods (Isaaks

and Srivastava, 1989; Barnes, 1964).

Two-stage (ANN + interpolation) models for predicting spatiotemporal variability of groundwater levels are conceptually intuitive and pragmatic. However, this approach has limited fidelity to the groundwater system it intends to model. The approach essentially decouples the temporal variations observed at the wells which is captured by fitting ANNs at individual wells and the regional-scale spatial variability which is encapsulated using a geostatistical method. The assumption that spatial and temporal

correlations can be decoupled, is not fully consistent with conditions in the field as temporal changes in neighboring wells can locally alter flow paths and affect water level forecasts at a given well.

As ANNs are universal approximators, it can be hypothesized that a single space-time formulation could conceivably be sufficient to model the regional-scale spatiotemporal water level dynamics. The integration of space-time dynamics into a single ANN architecture allows for simultaneous characterization of the coupled spatiotemporal dependencies using a parsimonious

model formulation. The primary goal of this study is to test this hypothesis by constructing a single space-time ANN model to capture the spatiotemporal variability in groundwater levels noted within a region. Such a model is referred to here as the "integrated space-time" or IST-ANN for brevity. The ANN model development is guided by the governing groundwater flow equation. Theoretical considerations for developing IST-ANN models are presented in the paper and the developed methodology is used to model regional groundwater flow in a portion of the Ogallala Aquifer in the Southern High Plains of Texas as an

illustrative case study.

## 2   IST-ANN methodology

### 2.1   Conceptualization of the IST-ANN architecture

Regional groundwater flow in an aquifer can be modeled using the governing groundwater flow equation which generically can be stated as:

$$S_s \frac{\partial H}{\partial t} = \frac{\partial}{\partial x}\left(K_{xx}\frac{\partial H}{\partial x}\right) + \frac{\partial}{\partial y}\left(K_{yy}\frac{\partial H}{\partial y}\right) + \frac{\partial}{\partial z}\left(K_{zz}\frac{\partial H}{\partial z}\right) + Q'_s \tag{1}$$

Where, H is the hydraulic head (L) and $S_s$ is the specific storage of the aquifer ($L^{-1}$); K is the hydraulic conductivity ($LT^{-1}$) and $Q'_s$ is the source/sink term ($T^{-1}$); x, y, and z are perpendicular to the cartesian coordinates and the x-axis oriented parallel to the direction of flow and t is the time. Source/Sink terms include recharge from precipitation, groundwater extractions from wells, exchange of water with surface water bodies (river, lakes) and other point, line and areal sources and sinks. The solution

of Eq. (1) subject to specified initial and boundary conditions yields values of hydraulic head H at locations denoted by (x, y, z) and at time (t). A first-order finite-difference formulation of Eq. (1) approximates the partial derivatives with difference





equations and the hydraulic head at any point in space (x, y, z) and time (t) can be expressed as:

$$H_{x,y,z,t} = H_{x,y,z,t-1} + \Delta H_{x,y,z,t} \tag{2}$$

$$\Delta H (x,y,z,t) \sim f\left( H_{x,y,z,t-k}, H_{x \pm \Delta x, y, z, t}, \ H_{x, y \pm \Delta y, z, t}, H_{x, y, z \pm \Delta z, t}, Q'_{x,y,z,t}, \ \theta_{x,y} \right) \tag{3}$$

As can be seen from Eq. (2), the hydraulic head at (x, y, z, t) is the sum of the hydraulic head measured at the same location at the previous time step (t-1) and the change in hydraulic head between time t and (t-1), which is denoted as $\Delta H_{x,y,z,t}$. This dependence of current water levels on values measured at previous time-steps denotes the persistence of the groundwater level time-series at a given location. Equation (3) also indicate that the change in water level is also a function of hydraulic heads at the neighboring cells and can be viewed as spatial dependence effects in the groundwater level time-series. Groundwater source

and sink terms (Q') at (x, y, z) and at time t also affect the hydraulic heads in the aquifer. These source/sink terms represent the external stresses imposed on the aquifer. Hydraulic heads also depend upon a vector of aquifer properties ($\theta$). These aquifer properties include hydraulic conductivity (k), specific storage ($S_s$), and aquifer thickness (H or b depending on whether the aquifer is confined or unconfined) all of which can vary in space. These terms highlight the dependence of groundwater level time-series on intrinsic aquifer characteristics. These aquifer characteristics typically exhibit spatial variability as well.

The governing groundwater flow equation is taken as the starting point for developing integrated space-time artificial neural network (IST-ANN) models. The hydraulic head (H) at a time (t) and at a well located at spatial location (x, y, z) is assumed to be a function of the lagged hydraulic head ($H_{x,y,z,t-k}$) at a given well. While higher order lags are usually neglected in standard finite difference groundwater flow simulators (Harbaugh, 2005), they have been noted to increase the accuracy of finite difference schemes (Mickens, 2005; Farthing et al., 2003).

Equation (3) indicates the groundwater level at a location (x, y, z, t) is a function of hydraulic head values at neighboring wells. In explicit finite difference schemes, these neighboring well heads at a time (t) are approximated by heads are previous lag(s) for mathematical simplicity. Such an assumption also becomes necessary for incorporating spatial dependence in feed-forward ANNs to avoid circular input-output relationships. Also, wells (grid locations where heads are computed) are seldom located at regular intervals in real-world monitoring networks (Uddameri and Andruss, 2014). As ANN models are data-driven,

the nearest neighbors will unlikely be equidistant at all locations of interest. The random well spacing is similar to using a non-uniform finite-difference grid for solving governing groundwater flow equation. In summary, governing groundwater flow equation inspires ANN models to include hydraulic heads from neighboring wells as they are likely to have a strong influence on forecasts and help capture the spatial dependence structure of the groundwater level time-series.

## 2.2  Input specification of regional ANN model

As in physical-based modeling, understanding the compatibility between the model and the system it seeks to represent is critical while developing ANN models. Choices for processes to consider, and how to represent them (i.e., parameterization) in the absence of direct measurements must be carefully ascertained (Oreskes et al., 1994). Data pertaining to source and





sink terms are seldom available in practical field applications (Russo and Lall, 2017) and are often estimated using surrogate measures even in physical groundwater modeling applications. This limitation also affects the development of regional-scale

ANN models. However, regional-scale ANN models allow one to directly incorporate surrogate information for sources and sinks as inputs and thus eliminate the need to convert them into flux terms needed in physically based models. This convenience removes the need for other empirical models and rules-of-thumb during the model parameterization process.

The specification of spatial variability of aquifer parameters (e.g., hydraulic conductivity) is another important consideration and also a major source of subjectivity when parameterizing physically based models (Gómez-Hernández and Gorelick, 1989).

Aquifer properties can be directly included in ANN models when available. However, as these properties exhibit spatial variability, it is also possible to capture their effects using location (x, y, z) coordinates. The use of location coordinates to capture spatial variability is similar to spatial regression approaches that have been used to regionalize groundwater level measurements (Nourani et al., 2008).

Based on the governing groundwater flow equation, a generic conceptualization of the IST-ANN model can be visualized

(see Fig. 1). While the groundwater flow equation provides a good starting point for identifying variables, the selection of certain features (e.g., how many spatiotemporal lags to consider) cannot be directly ascertained. In addition, when aquifer stressors (e.g., pumping) are not directly measured, the selection of an appropriate surrogate cannot be based strictly on physical considerations, especially when more than one surrogate can be used to model the phenomenon of interest. Therefore, as a first step, it is best to cast a wide net and create a comprehensive database of potential inputs (directly measured values and suitable

surrogates) for constructing ANN models. Statistical considerations and machine learning based parameter selection criteria have been proposed for dimensionality reduction and selection of inputs (Wei et al., 2015; Bi, 2012; Kursa et al., 2010; Kleijnen and Helton, 1999). These approaches can be used to modify and refine physics-inspired specification of ANN architectures in practical modeling applications.

## 2.3 Data preprocessing to refine candidate input parameters

Highly correlated input variables share a large amount of common information and therefore hamper the learning ability of ANNs (Ahire, 2018). Correlated inputs arise when more than one surrogate can be identified to represent a physical process in a regional-scale ANN. Similarly, lagged variables may also be strongly correlated when the underlying physical processes controlling these variables exhibit persistence. Input multicollinearity can be used as a criterion for the elimination of certain inputs. The mutual information (MI) is an entropy-based metric that measures the amount of information one random

variable has about the other (Bennasar et al., 2015; Vergara and Estévez, 2014). MI can be used to not only rank variables according to their importance, but also select among competing surrogates to represent a process (i.e., variables likely to cause multicollinearity issues). MI can be calculated using Eq. (4), where p denotes the probability density and $x_i$ and $x_j$ are two (competing) model inputs.

$$MI(x_i, x_j) = \sum \sum p(x_i, x_j) \log \left( \frac{p(x_i, x_j)}{p(x_i) p(x_j)} \right) \Delta x_i \Delta x_j \qquad (4)$$





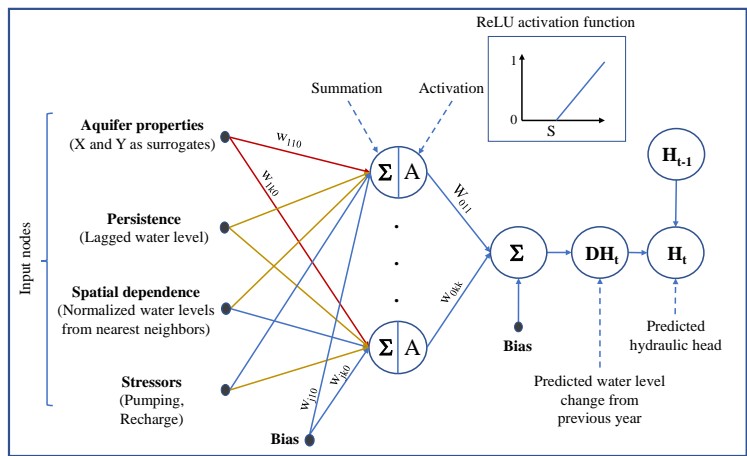

**Figure 1.** A generic IST-ANN model architecture

Random Forests (Breiman, 2001), RF, is another machine learning technique that can be used to ascertain variable importance. It has been widely used in several studies as a pre-processing step to identify suitable input variables (Grömping, 2009; Siroky, 2009; Strobl et al., 2008, 2007). Briefly, RF algorithms construct several decision trees by – 1) using a random set of observations and 2) a random subset of candidate variables. The observations not used to build the decision tree are referred to as the out of the bag (OOB) samples. The accuracy of a RF model is proportional to its ability to correctly predict OOB samples (as measured using mean square OOB error, referred to here as OOBMSE). If a candidate input variable ($X_j$) has no influence on the output, then random permutation of $X_j$ in OOB samples does not alter the OOB mean square error (OOBMSE) appreciably (i.e., permuted OOBMSE $\sim$ unpermuted OOBMSE). Variables that have the highest influence cause the greatest increase in the MSE upon permutation. This concept is used to compute the average change in OOB error associated with each input variable ($X_j$) of interest over all trees in the forest and rank inputs according to their importance (Ishwaran et al., 2008).

   Variable selection approaches are non-unique and different approaches can point to different inputs. As such, the use of both MI and RF based methods is recommended and adopted in this study. A variable is clearly important if it ranks highly in both techniques and has a well-defined physical meaning. A combination of statistical and physical considerations, theoretical underpinnings and alternative model conceptualizations can be used to deal with those variables that rank high under some techniques but not in others.

## 2.4 IST-ANN model specification

The number of hidden layers play a key role in defining the architecture of ANNs and therefore its specification is an important factor in building these models. In the spirit of model parsimony, a single hidden layer ANN was selected to model regional-scale space-time predictions as the interpretation of this architecture is straightforward and sufficient to provide universal





approximation capabilities (Hornik et al., 1989). The selection of the activation function, which transforms the weighted sum
of the inputs entering the hidden node into an output node is another important decision when configuring ANNs. The rectified
linear unit (ReLU) is a non-saturated activation function which assumes a value of zero when the weighted summation at the
hidden node is less than zero and is equal to the weighted summation otherwise (g(z) = max(0, z)). ReLU was adopted here (see
Fig. 1) as it is known to provide faster convergence with gradient descent algorithms than conventional saturating activation
functions such as sigmoid and tanh that have been used in previous groundwater applications of ANNs (Djurovic et al., 2015;
Tao et al., 2015; Uddameri, 2007; Nayak et al., 2006).

Overfitting is a major concern in ANN modeling and occurs when the model memorizes the training dataset but has poor
generalizability (Uddameri, 2007). As overfitting is caused due to the presence of many hidden nodes (some which only learn
the noise in the data), the L2 regularization term was added to the objective function to keep the number of hidden nodes to a
minimum. As can be seen from Eq. (5), the regularization term increases the objective function when more nodes are added to
minimize the mean square error (MSE).

$$f(x) = MSE + \lambda \sum_{i}^{k} w_i^2 \tag{5}$$

The weighting factor, $\lambda$, in Eq. (5) can be treated as a hyper-parameter and estimated as part of the model calibration process.

The adaptive moment or ADAM stochastic gradient descent method (Kingma and Ba, 2014) was used for the minimization
of the objective function (see Eq. (5)). The ADAM method computes different learning rates for individual parameters which
are computed from the bias-corrected first- and second- moments of the gradients. The method offers several advantages in that
the parameter estimates are invariant to the rescaling of the gradients, and the method can effectively deal with sparse gradients
and non-stationary objective functions. The ADAM algorithm adds three additional hyper-parameters ($\alpha, \beta_1, \beta_2$) that control
the step-size and exponential decay rates. Kingma and Ba (2014) provided good default values for these hyper-parameters for
use with machine learning optimization problems and were noted to be adequate in this study.

A K-fold cross-validation (K = 5) approach was adapted here to further minimize any overfitting concerns. K-fold cross-
validation involves randomly splitting the training dataset into K equal folds, train a model over (K-1) folds and test its adequacy
to make predictions using the unused ($K^{th}$) subset (i.e., validation fold). The procedure repeats K times and at each time a
unique fold is treated as a validation subset. The final skill (fitness) score is computed by averaging the errors over all K rounds.
The training algorithm was also run in the mini-batch mode, where the parameters get updated over a few randomly selected
samples rather than the whole dataset as this approach is known to reduce variance in parameter updates and lead to stable
convergence (LeCun et al., 2012).





## 3 Illustrative case-study

### 3.1 Study area

The Ogallala Aquifer underlying the High Plains Underground Water Conservation District #1 (HPWD) was selected to
illustrate the development of the proposed IST-ANN model (see Fig. 2a). The study area is a semi-arid region with cool winters
and hot summers and spans over 30,691 $km^2$. The topography of the region is characterized by rolling plains and land surface
elevations range from 1353 m in northwestern part to 810 m in southeastern portions of the study area (Gutentag et al., 1984).
The average annual precipitation is around 480 mm, of which 80 % occurs from April to September (PRISM Climate Group,
2019) during which most agricultural crops are grown. Seasonal precipitation is often insufficient to meet crop water demands
and groundwater from the underlying Ogallala Aquifer is heavily used for irrigation.

The study area is a groundwater dependent agricultural economy (Uddameri and Reible, 2018) and accounts for over 20 %
of the nation's cotton production (USDA-NASS, 2012). Over-exploitation of the aquifer has led to severe water table declines
in the region (Scanlon et al., 2012). The food, fiber, and livestock produced in the region are exported across the world and
therefore the depletion of groundwater in this agriculturally intensive region is not only important to sustain the local economy
but also has severe consequences pertaining to global food security (Marston et al., 2015). The HPWD was created with
the mission of conserving and protecting groundwater resources while ensuring its availability and use for the economic
development of the region. As part of its mission, the HPWD has conducted annual groundwater level monitoring campaigns
since its creation in 1951. Water levels are collected in the winter between December–March as groundwater production during
this period is low and wells exhibit recovery from summer (irrigation) production back to near static conditions. The data from
this sustained long-term monitoring effort was exploited to develop the IST-ANN model in this study.

### 3.2 Data compilation

Groundwater level measurements and well information were compiled from the Texas Water Development Board database
(TWDB, 2019). A total of 149 wells (see Fig. 2a) having at least 50 years of annual water level measurements between 1955–
2015 were selected for this study. The Kalman filtering approach was used to impute the missing data to create continuous
annual groundwater level time-series data for the period of 1955–2015 (Herrera and Pinder, 2005). Groundwater level changes
were computed on an annual basis. For most wells, there was only one reported water level value each year. An average value
of the groundwater level was used as a representative value when more than one measurement was made during the winter
months. Following (McGuire, 2017) water level measurements made during summer and fall, if any, were discarded as they do
not represent near static conditions.

Groundwater levels in unconfined aquifers are often a subdued replica of the topography, especially when the recharge is
low (Gleeson et al., 2012). As the elevation of the region slopes in the NW-SE direction, the coordinates (X and Y measured
in Albers Equal Area projection) of the well were taken as surrogates for the topography. In addition, the aquifer bottom
and other aquifer properties exhibit spatial variability. Therefore, the coordinates (X, Y) also serve as indirect surrogates for
hydrogeological variability.





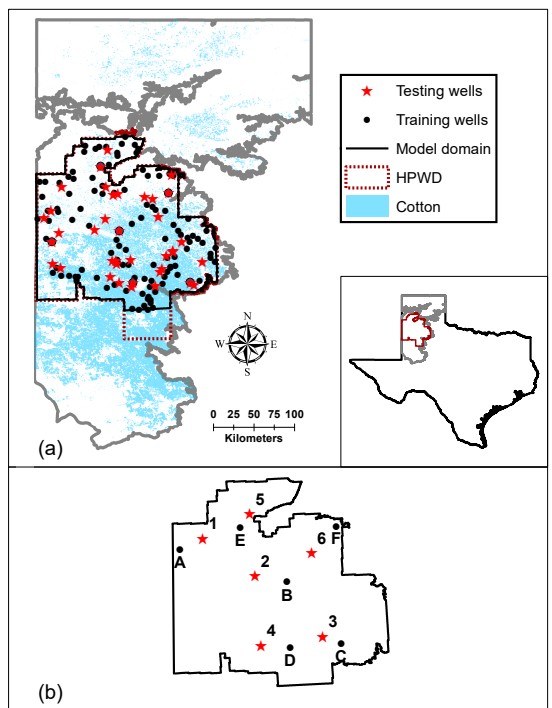

**Figure 2.** High Plains Groundwater Conservation District (HPWD)

As water tables are deep in the region, recharge to the aquifer is practically negligible (Scanlon et al., 2012). Groundwater discharges to streams and rivers are virtually non-existent due to the lack of hydraulic connection between surface water bodies and the aquifer within the region (Uddameri et al., 2017). Pumping is the major source of groundwater discharge. Historical groundwater pumping data were not available in the region and this is acknowledged as a major problem in developing regional-scale models in many agricultural areas (Russo and Lall, 2017). However, as groundwater production is largely for agricultural

purposes, the correlation between climate state (drought) indicators and pumping is noted to be strong (Whittemore et al., 2016) and therefore a useful surrogate for representing pumping in the ANN model.

    The Standardized Precipitation Evapotranspiration Index (SPEI) (Vicente-Serrano et al., 2010) with an accumulation period of three months (SPEI-3) was summed annually (Jan–Dec) and used as a surrogate for total annual pumping. The three-month SPEI is known to correlate well with agricultural water demands in the Great Plains region to which the study area

belongs (Zambreski et al., 2018) and as such adopted here. The SPEI-3 values were computed using monthly precipitation and temperature data obtained from PRISM (PRISM Climate Group, 2019) and making use of the Thornthwaite equation for the potential Evapotranspiration (Thornthwaite and Mather, 1957). Preliminary comparisons of PRISM data with those measured at meteorological stations indicated the reasonableness of the gridded dataset in providing representative precipitation and



temperature data within the study area. This dataset provided good regional-scale coverage and thereby removing the need to
impute missing temporal data at individual weather stations and also helped minimize errors associated with the regionalization
of station-level data.

Water levels at each well , GWL(t), in the year (t) (i.e., the state variable) along with lagged water levels at three previous
years labeled as GWL(t-1), GWL(t-2) and GWL(t-3) were compiled into the initial master database. As monitoring wells are
randomly scattered in space (Fig. 2a), the neighboring wells were not equidistant. Therefore, an inverse distance normalized
lag-1 hydraulic head was used for the neighboring wells (i.e., lag-1 hydraulic head at the neighboring well divided by the
distance to the neighboring well) to normalize the effects of non-uniform well spacing. This parameter called normalized
neighboring head was estimated for four nearest neighboring wells and labeled as N-1(t-1), N-2(t-1), N-3(t-1), N-4(t-1) were
also compiled into the master database. SPEI values (a surrogate for water demands and pumping) for the current year , SPEI(t),
as well as the previous two years — SPEI(t-1) and SPEI(t-2) — were also added to the database. As the focus was on modeling
annual water level time-series, three temporal lags and four nearest neighbors were deemed sufficient based on autocorrelation
analysis. Higher order lags were noted to have very low autocorrelation. The initial feature (master) database had 116,220
records (149 wells $\times$ 60 years $\times$ [12 features + 1 state variable]).

### 3.3 Training, validation, testing and forecasting datasets

The 149 monitoring wells were randomly split into 111 training wells (75 %) and 38 (25 %) testing wells (See Fig. 2a). The
water levels observed in the year 1955 was used as the initial condition. The time-period 1956–2015 was split into training and
testing (1956–2008) and forecasting (2009–2015) time periods. As shown in Table 1, water levels for the period 1956–2008 at
the 111 training wells were used to obtain the connection weights and values for hyperparameters (i.e., the minimum number
of hidden nodes, optimal objective function weighting factor, $\lambda$, and learning rate). The water level time-series data from 1956–
2008 at the 38 testing wells were used to independently evaluate the prediction abilities of the model (testing error). The water
level time-series from 2009–2015 at all 149 wells were also used to evaluate the forecasting abilities of the models.

The one-step ahead forecasting approach was the primary focus of this study. The training, testing and forecasting abilities
were therefore evaluated in this mode. In other words, forecasts were made for time-step (t) using observed data from previous
(t-1) time-step. However, the model can also be used to make multi-step ahead predictions using a recursive approach. The
performance of the models was evaluated using a suite of methods which included visual observations of well hydrographs,
computation of individual well level performance measures, spatial mapping of error metrics and comparison of observed and
simulated hydraulic heads at various time periods. In particular, the mean absolute error (mean non-directional error), bias
(mean directional error), and correlation coefficient were used to quantify the total error, systematic bias and the strength of
the relationship between observed and modeled values. In addition, the Kling-Gupta Efficiency (KGE) metric (Kling et al.,
2012) was used to evaluate the overall predictive performance considering correlation, bias, and variability. These measures
are summarized in Table 1.

The IST-ANN modeling was carried out using the Keras open-source neural network library (Chollet et al., 2015) with the
TensorFlow (Abadi et al., 2016) computational engine in the backend in Python 3.7.3. In addition, libraries for performing



**Table 1.** Performance error metrics used

| Parameter | Formula | Remarks |
| --- | --- | --- |
| Mean Absolute Error (MAE) | $\frac{1}{N}\sum_{i=1}^{N}\lvert S_i - O_i\rvert$ | Total error without consideration of direction. Overall model performance assessment. S is simulated values; O is observed values and N is total number of data points. |
| Bias | $\frac{1}{N}\sum_{i=1}^{N}(S_i - O_i)$ | Total error considering the direction of the prediction. A value of zero implies random errors, a non-zero value points to a net systematic overprediction or underprediction (i.e., net bias) of the model. S, O, and N as defined above. |
| Correlation Coefficient | $\dfrac{\sum_{i=1}^{N}(S_i - \bar{S})(O_i - \bar{O})}{\sqrt{\sum_{i=1}^{N}(S_i - \bar{S})^2 \sum_{i=1}^{N}(O_i - \bar{O})^2}}$ | A strength of the monotonic relationship between observations and predictions. A value of +1 indicates a perfect model fit. A value of 0 indicates a poor fit. Negative values imply the model is predicting the opposite of what is noted in the observations. S, O, N as defined above. Overbar values indicate mean values of the corresponding variables. |
| Kling-Gupta Efficiency | $1 - \sqrt{(r-1)^2 + (\alpha-1)^2 + (\beta-1)^2}$ | An overall error measure considering correlation, bias, and variability. The value of unity implies a perfect model fit. r is the Pearson Product Moment correlation; $\alpha$ is bias term and $\beta$ is the variability term. See Kling et al. (2012) for further details. |

Random Forest-based feature selection (Strobl et al., 2007), mutual information calculations (Meyer, 2014) and computation of model performance measures (Zambrano-Bigiarini, 2014) were carried out by developing custom scripts in R programming environment and have been made publicly available (Ghaseminejad and Uddameri, 2020).

## 4 Results and discussion

### 4.1 Input selection

The Mutual Information (MI) criterion and the Random Forest-based variable importance metric (relative importance score) along with physical considerations were used to reduce the features (inputs) of the IST-ANN model. Figure 3 indicates that the coordinates (X and Y) were relatively important variables with both selection criteria and as such were retained in the final model. The X and Y coordinates were used as surrogates to model the observed variability in the model domain (Gutentag et al., 1984) and as such their relative importance was to be expected. The Y-coordinate was seen to be more important than the X due to a greater variability along the North-South (N-S) axis more so than the East-West (E-W) axis. There is a considerable



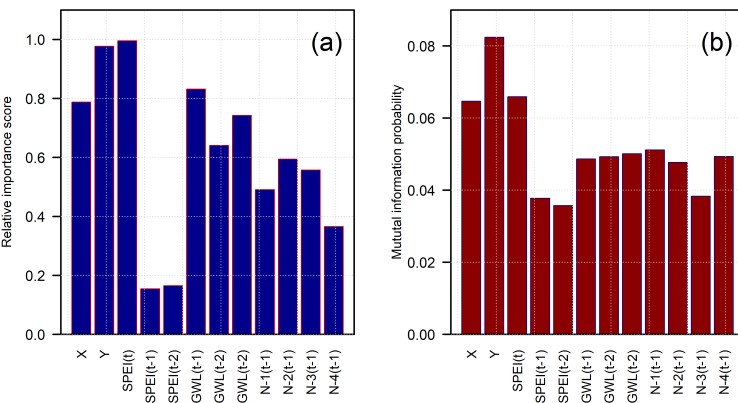

**Figure 3.** Parameter relative importance analysis for different inputs

climate gradient along the N-S axis (cooler in the north and warmer in the south) which controls the crops grown and therefore

affects the amount of groundwater extraction more so than the topographical gradient found along the E-W direction. Also, the aquifer saturated thickness exhibits a general N-S variability (McGuire, 2017) and this, in turn, affects the transmissivity of the aquifer within the study area.

Figure 3 also indicates that lags of SPEI have relatively low importance compared to SPEI values in the current year. This result is again to be expected as water levels are most affected by groundwater pumping in the immediate past as limited (or

no) production during the winter months allows the water table to recover prior to being stressed again in summer months. As such, SPEI in the current year was chosen as a model input and lagged values of SPEI were removed from further consideration in the interest of parsimony.

The lagged water levels at a well represent the memory of the aquifer system to previous stresses (persistence). The lag-1 (previous year) groundwater level was deemed the most important as per the Random Forest model. However, the MI criterion

indicated that roughly similar information was shared between the water level in the current year (t) and three previous years (t-1, t-2, and t-3). Finite-difference groundwater flow simulators such as MODFLOW (Harbaugh, 2005) often produce excellent results considering only lag-1 water levels (arising from first-order differencing of the temporal derivatives). Also, as water tables are noted to recover during winters, the lag-1 water level should incorporate the effects of previous stresses on the aquifer. Therefore, only the lag-1 water level was retained as a model input. In a similar vein, the normalized water levels from

four nearest neighbors were retained based on MI criterion and are akin to the central differencing schemes often employed to represent spatial derivatives in finite-difference groundwater flow simulators (spatial variability terms). The final IST-ANN model therefore comprised of eight inputs (X, Y, SPEI(t), GWL(t-1), N-1(t-1), N-2(t-1), N-3(t-1) and, N-4(t-1) in addition to the bias terms.



## 4.2   Training of ANN models

The performance of the IST-ANN model was a function of the regularization weight, the number of hidden nodes, and the initial
learning rate (see Fig. A1 in the appendix). Setting learning rate = 0.01, $\lambda$ = 0.005 and, number of hidden nodes = 12 yielded
the lowest mean absolute error over the cross-validation. The low value of regularization weight indicates that overfitting was
not of concern here. The 12 hidden nodes is well within the recommended range of (2–17) hidden nodes for an 8 input ANN
(Maier and Dandy, 2001) and as such the final architecture was deemed reasonable for the purposes of the study.

The stochastic gradient descent calibration procedure of the IST-ANN model resulted in excellent results across the study
area. The mean absolute prediction error was generally below 0.75 m except at a few wells which had much significant local
influences. The calibration model exhibited low bias and predictions were generally between ± 0.2 m for over 67 % of all
training wells. In general, the model had exhibited a greater propensity to overestimate the hydraulic heads ( 26 % of the wells)
than underestimate the head ( 8 % of training wells). This overestimation is seen along the NW-SE direction within the study

area (see Fig. 4b), which represent areas of intensive irrigated agricultural activities. However, the maximum net bias was 0.5
m and as such the calibrated predictions are not too far off from the observed values. In general, an overestimation of hydraulic
heads implies an underestimation of the groundwater depletion and points to the limitation of using SPEI as a surrogate for
groundwater pumping. While SPEI provides a general trend of the climate state, irrigation also depends upon the timing of the
rainfall events and days with high temperatures. Substantial irrigation may be necessary even during normal and wet years, if

the rainfall events do not coincide with critical plant growth stages or the region experiences higher temperatures during rapid
plant growth stages which considerably increase the water demand (Uddameri et al., 2017).

The correlation coefficient was > 0.8 for 95 % of the training well and the overall KGE was greater than 0.8 for 94 % of
the calibration well indicating the model was able to capture the overall trends in the data and model predictions generally
were not affected by bias and variability as measured using KGE. Lower KGE values generally corresponded to areas with

high localized groundwater production (e.g., in Bailey county where the city of Lubbock, pop.~250,000, the largest city in
the region has a well field) and in Crosby, Floyd and Lubbock counties where municipal pumping well fields for smaller
cities (pop. < 25,000) exist. While urban water demands are influenced by meteorological conditions incorporated within SPEI
(Adamowski et al., 2013), they are also influenced by other factors such as conservation programs (e.g., lawn water restrictions)
and cost. Municipal water demands were not included in the present modeling study as urban areas represent less than 2 %

of the study area. However, the results indicate that efforts to capture municipal water demands may locally improve model
calibrations. All in all, the calibration of the IST-ANN model can be deemed suitable for regional-scale applications based on
the spatial evaluation of time-aggregated performance metrics shown in Fig. 4.

Figure 5 presents representative observed and modeled well hydrographs at selected calibration wells (see Fig. 2b). The
selection of these representative wells was to visualize temporal water level prediction patterns across the region especially at

those wells that had relatively higher mean absolute errors (MAE). Figure 5 indicates that the IST-ANN model is capable of
capturing both the long-term declining trends as well as short-term volatility caused by anomalous weather events that allowed
for higher than normal water level recovery. The results from the well hydrographs in conjunction with the time-aggregated

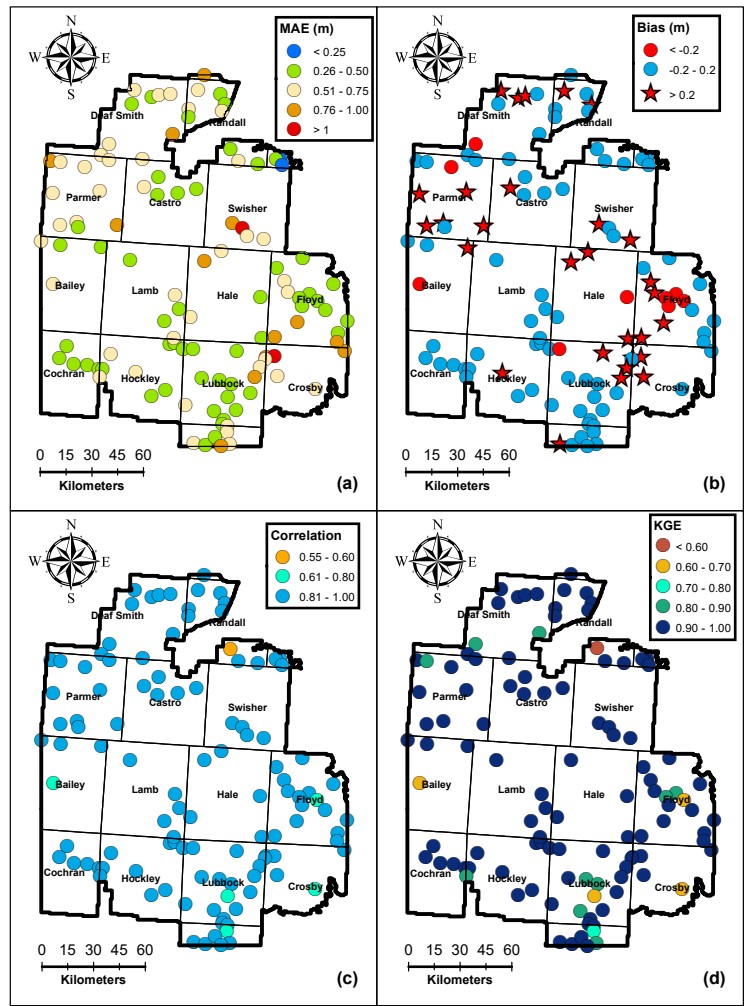

**Figure 4.** Spatial variability of performance measures at 119 training wells for the calibration period 1956–2008

performance metrics indicate that the IST-ANN could learn both the spatial and temporal patterns in data using a relatively small set of input parameters. While obtaining good calibration is a necessary first step, independent testing of the model at
wells not used for calibration is critical to gain confidence in the model and ensure the model has not simply memorized the training dataset (i.e., overfitting concerns).

## 4.3 Independent testing of IST-ANN model

Independent testing of the model performance was carried out 38 wells that were not used for calibration. The time-aggregated performance metrics depicted in Fig. 6 indicate that the model was able to predict groundwater levels at these testing wells with
a high degree of accuracy. The mean absolute error for testing wells was less than 0.75 m for 77 % of the wells and less than



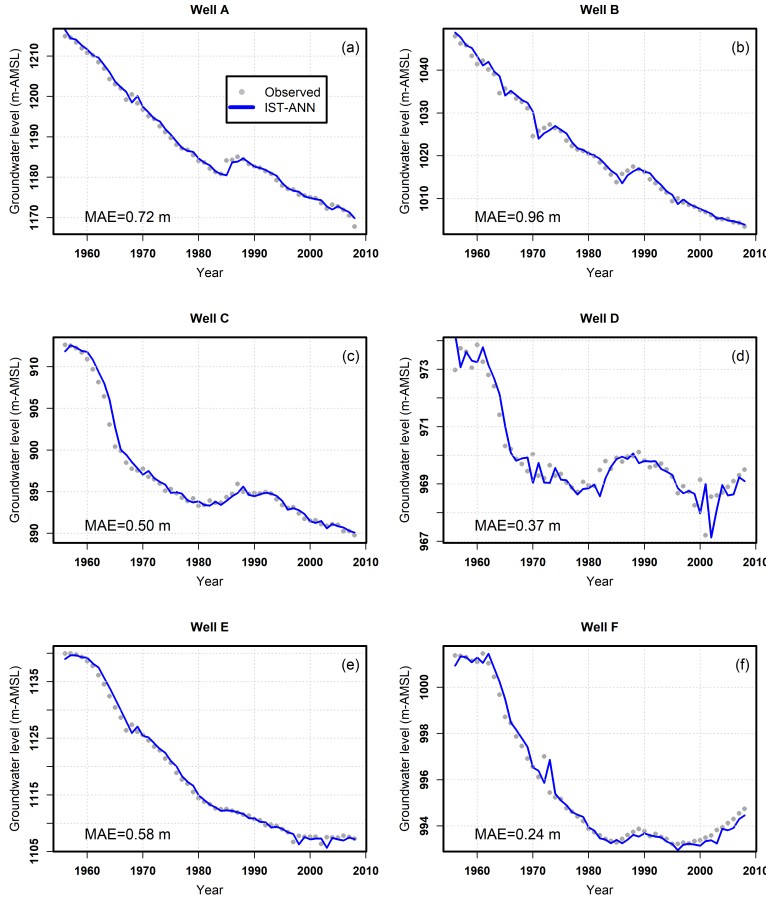

**Figure 5.** Well hydrographs at select training wells during the calibration period 1956–2008 (also see Fig. 2b for well locations)

1.0 m at 93 % of the wells. The maximum mean absolute error was 1.22 m for the testing dataset which compares favorably with the maximum mean absolute error of 1.14 m for the training dataset. The net bias depicted in Fig. 6b shows similar trends as the training dataset with a general overestimation of the hydraulic heads along the NW-SE direction. The maximum net bias was 0.52 m in the testing dataset which is again similar to that observed in the training dataset (0.50 m). The correlation
coefficient as well as KGE were greater than 0.8 at over 95 % of the testing wells. These results indicate that the calibrated IST-ANN models were able to predict the overall observed behavior in independent wells during the 1956–2008 calibration period. These results confirm that overfitting was not an issue and highlight the ability of the regularization approach to keep the number of hidden nodes to a minimum and help the IST-ANN model focus on its generalization abilities.

In addition to time-aggregated performance metrics, the ability of the IST-ANN model to capture temporal trends at individ-
ual wells were also assessed. Figure 7 depicts the well hydrographs at six representative wells having different mean absolute errors and scattered across the study area. The results again indicate that the IST-ANN model is able to adequately capture the



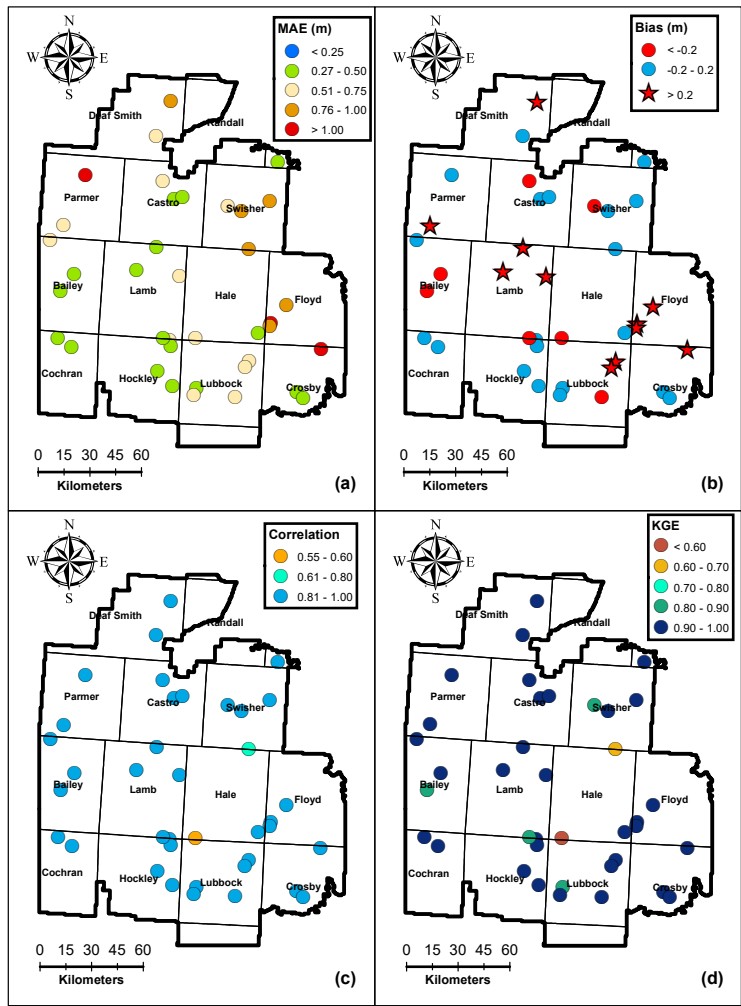

**Figure 6.** Model performance metrics for the testing dataset (38 wells) for the period of 1956–2008

observed long-term trends in the data. Short-term volatility is pronounced in some wells (see Fig. 7a, d–f) which the IST-ANN model captures well. These results indicate that the IST-ANN not only performs well over a suite of time-aggregated metrics but also is able to capture nuances of water table dynamics noted across the study area. The independent testing of the model adds confidence to the model predictions and confirms the generalization capabilities of the IST-ANN modeling approach.

## 4.4 Evaluation of forecasting capabilities of the IST-ANN model under extreme climate conditions

While the IST-ANN model was seen to perform well on historical datasets (1956–2008), the performance of the model to forecast recent water levels is of interest. The period of 2009–2015 was used to assess the forecasting abilities of the model. The forecasting time-period represents a time-capsule of extreme climate variability as it was characterized by one of the



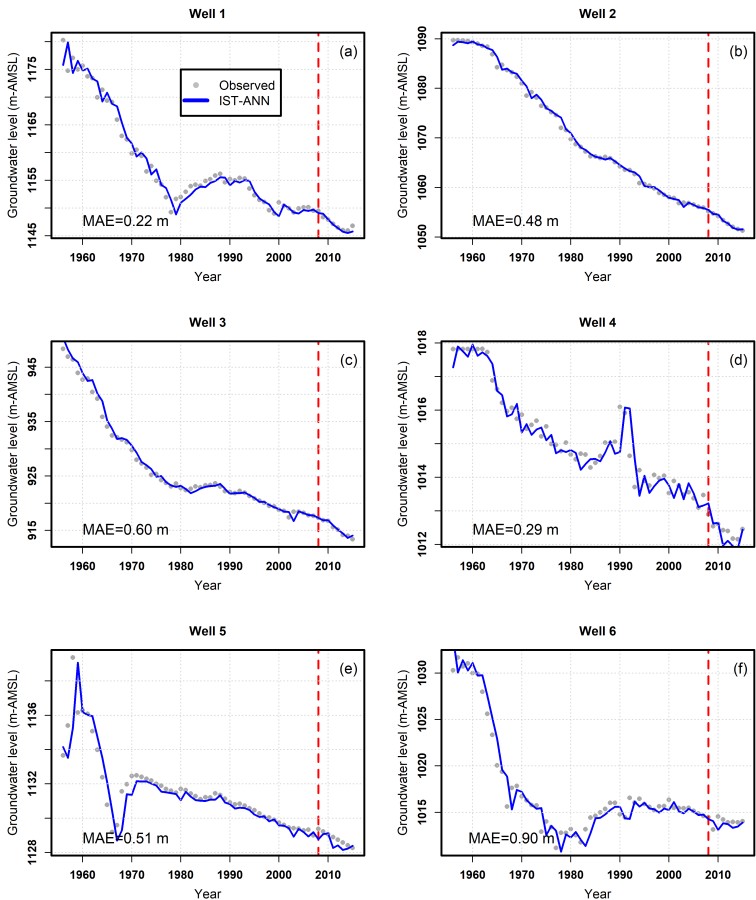

**Figure 7.** Well hydrographs for representative testing Wells within the study area (see Fig. 2b for locations of these wells; The right-hand side of the model denotes forecasting period)

worst droughts seen in the area, including the worst single year drought in the recorded history of Texas (year 2011) as well as one of the wettest year (year 2015) recorded in the region. As shown in Fig. 8, the model had to forecast in a range of hydrometeorological conditions that were not represented during the calibration process. As such, the forecasting time-period is largely a measure of the extrapolation capabilities of the model.

From a theoretical standpoint, ANNs are not designed to perform extrapolation outside their calibration range (Principe et al.,
2000). However, from an application perspective, climate variability is becoming more pervasive and extreme (Thornton et al., 2014; Gutzler and Robbins, 2011). In all likelihood, groundwater models have to be used to make forecasts under hydroclimatic conditions that are very different from those observed during their calibration. Therefore, the ability of data-driven models (which invariably have to be calibrated using long-term historical data) to make forecasts under different hydrometeorological conditions serves as an important check on their suitability in practical groundwater management applications.



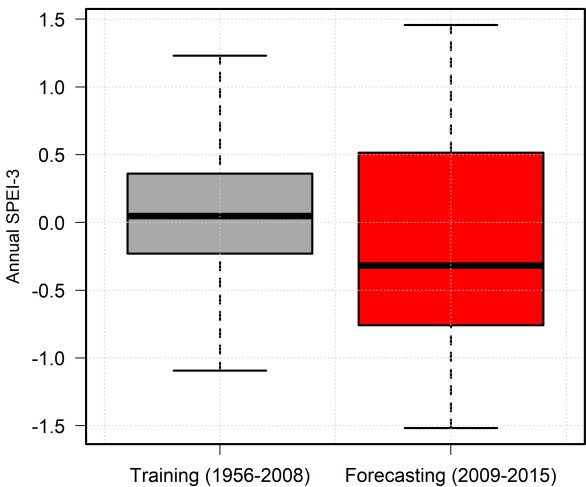

**Figure 8.** Variability of hydrometeorological conditions during training and forecasting period

Model forecasts for the years 2009–2015 were developed at all 149 wells as data from this period was not used for calibration purposes. Figure 9 depicts the performance metrics of the IST-ANN model over the forecasting time-period. The mean absolute error (MAE) was less than 0.75 m at 92 % of the wells and less than 1.0 m at 98 % of the wells. The maximum MAE was 1.24 m which is comparable to that observed during the testing phase (1.22 m). However, the net bias was within ± 0.2 m at 48 % of the wells compared to 53 % in the testing phase. In particular, the maximum observed overprediction (1.04 m) and

underprediction (-0.97 m) were much higher than those noted in the testing period (0.52 m and -0.43 m). In addition, the model underpredicted at 42 % of the wells and overpredicted at 11 %. The correlation coefficient was higher than 0.8 in over half of the wells and greater than 0.6 in about three-fourth of the monitoring wells. The KGE values were greater than 0.8 in only 35 % wells but values were greater than -0.41 in 90 % of the wells indicating that the model forecasts are significantly better than the mean values (Knoben et al., 2019). The well hydrographs in Fig. 7 for the forecasting time-period indicate that the

IST-ANN model was typically unable to capture the noted variability in the period but was able to discern general trends in well hydrographs. The variability ratio (Kling et al., 2012) for training, testing, and forecasting datasets is depicted in Fig. 10 and highlights the high degree of variability noted during the forecasting period.

    Farmer preferences tend to be heterogeneous within a region (Adhikari et al., 2010) and this heterogeneity was further exacerbated during the prolonged, severe drought experienced during the forecasting period. Some producers in the region chose

to increase their groundwater pumping or switched from dryland to irrigated agriculture. Other farmers adopted conservation practices which led to reduced groundwater use. These heterogeneous farmer preferences and their adaptation to prolonged drought and extreme climate explain the observed variability in Fig. 10.



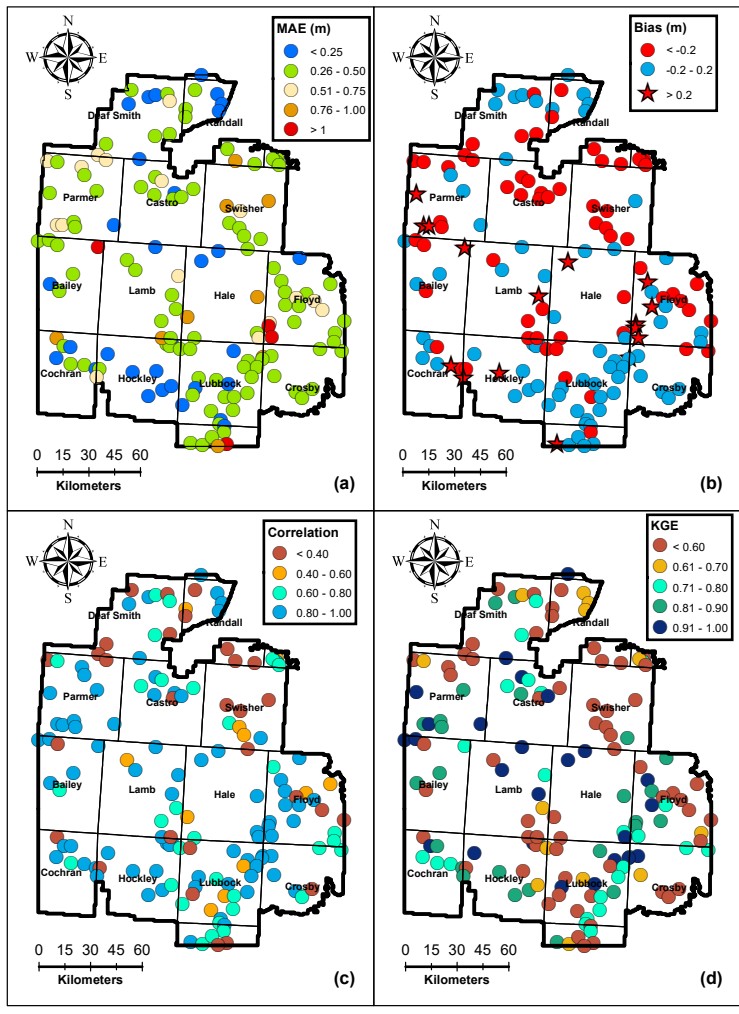

**Figure 9.** Time aggregated performance metrics for the validation period (2009–2015) for one-step ahead forecasting at 149 monitoring wells

There were no variables in the model to capture these preferences of the groundwater users in the region. The spatial variability in SPEI (surrogate for pumping) was considerably small as all of the study area experienced severe drought conditions and this parameter was insufficient to capture the observed variability in pumping. The difference in observed and simulated variance played a critical role in reducing the correlation coefficient and KGE metrics during the forecasting period.

As climatic conditions play a critical role in defining groundwater extractions, spatial variability of forecasting errors was assessed by comparing with observed water levels in 2011 (worst drought year in the recorded history of Texas; mean SPEI-3 ~ -1.5), 2014 (a relatively normal year; mean SPEI-3 ~ 0), and 2015 (a wet year; mean SPEI-3 ~ +1.3). The results depicted in Fig. 11 show that the fraction of the study area where the predicted drawdown error was less than 1 m was highest under



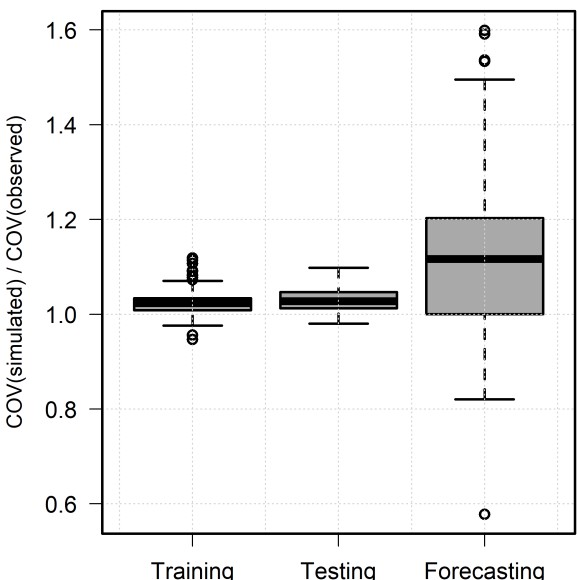

**Figure 10.** Variability ratio for training, testing, and forecasting datasets

normal climatic conditions and deviated under both dry and wet conditions. However, the deviations were relatively small under wet conditions than under dry conditions. The absolute error was less than 2 m for 88 % of the study area in 2011 (dry year) but was 92 % and 94 % in normal (2014) and wet (2015) years. This result highlights that in addition to the volume of the net rainfall (that is captured using SPEI-3) the timing of the rainfall events also play an important role. Even in relatively

wet years, rainfall may not occur during critical crop growth periods and warrant significant levels of irrigation. While SPEI correlates strongly with agricultural conditions, it is not an agricultural drought indicator per se. Agricultural drought indices, especially those using soil moisture measurements, could serve as better surrogates to capture irrigation dynamics. However, ground-truthing soil moisture-based agriculture drought indicators is often challenging due to paucity of data , and these soil moisture indices do not include irrigation water applications (Tsige et al., 2019). Therefore, these indicators may not fully

substitute for the lack of groundwater pumping data.

## 4.5 Input parameter sensitivity

The relative impact of input parameters on the output was explored by looking at connection weights of the IST-ANN model (Olden et al., 2004). The sensitivity of the model output to various input values are depicted in Fig. 12 (also see Appendix B for relative important index calculation). The predicted groundwater level changes were most sensitive to SPEI as irrigated

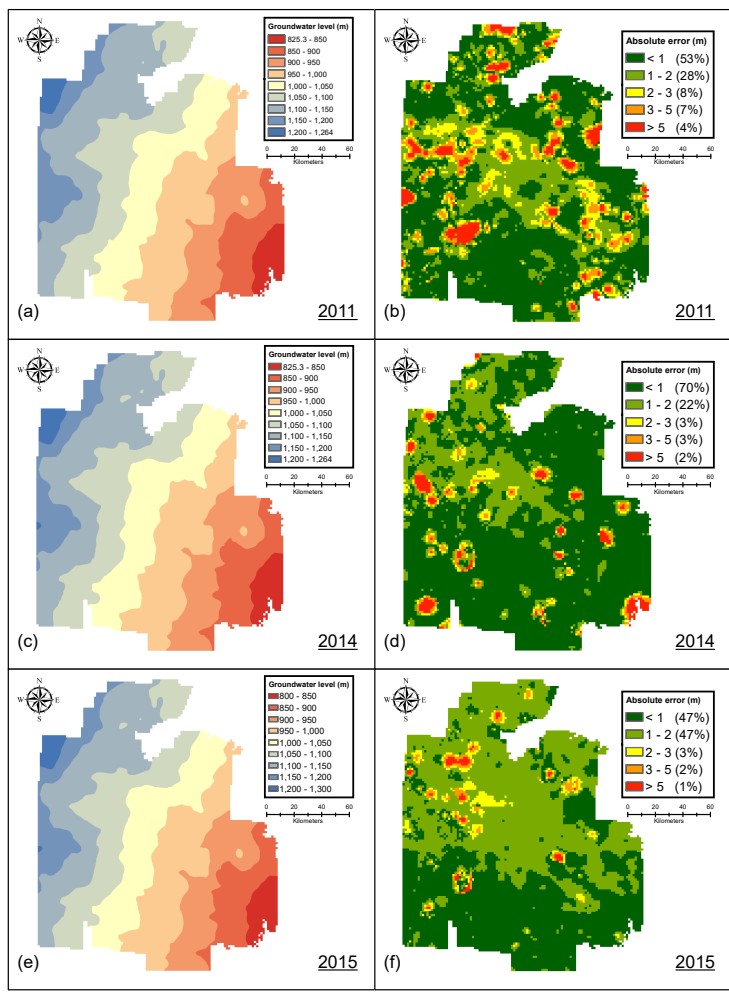

**Figure 11.** One-step ahead forecasting errors during dry (2011), normal (2014), and wet (2015) years

agriculture is the primary user of groundwater in the study area. The well coordinates were used as a surrogate to capture the variability of the aquifer properties. The saturated thickness in the study area has a distinct north-south variability (McGuire, 2017). This variability in turn affects the aquifer transmissivity a key hydrogeological parameter that affects hydraulic heads and explains the high sensitivity of the model output to the Y coordinate.

The sensitivity analysis also suggests that the normalized hydraulic heads at neighboring wells (spatial dependence terms)
affected the model predictions followed by the lagged groundwater level. The predicted groundwater level at a time (t) at a given well depends upon how quickly water can flow into the well. Both the groundwater level at the neighboring well N-1(t-1) and GWL(t-1) are necessary to compute the hydraulic gradient, which determines the rate of water movement to a well. The predicted groundwater level was also seen to be somewhat sensitive to the normalized hydraulic head at the fourth nearest well





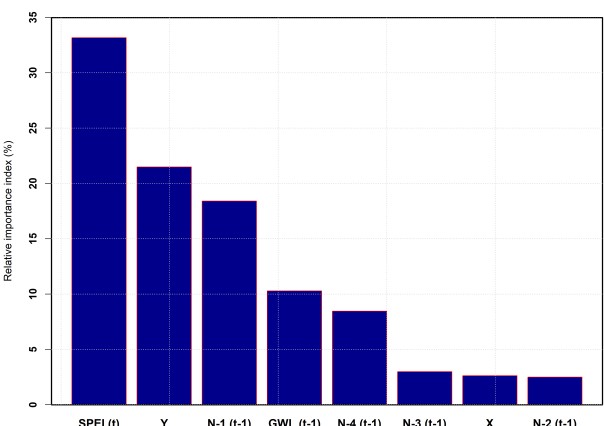

**Figure 12.** Relative importance of model inputs on predicting groundwater levels

(N-4(t-1)). The fourth nearest well was typically located along the Y-axis and this sensitivity denotes the anisotropic nature of
groundwater flow at least in some portions of the study area (Gutentag et al., 1984).

## 5    Summary and conclusions

The primary goal of this study was to evaluate the hypothesis that a parsimonious ANN model can be constructed to model
regional-scale aquifer dynamics in space and time. An integrated space-time artificial neural network model (IST-ANN) was
specified drawing inspiration from the governing groundwater flow equation. The lack of data to directly include pertinent
hydrogeological processes is a challenge in practical groundwater modeling studies. Machine learning based feature selection
algorithms were therefore used to identify a suitable set of inputs to parameterize the model. Recent advances in ANN model
calibration, namely—regularization, adaptive moment stochastic gradient algorithm along with 5-fold cross-validation were
used to minimize model overfitting. A 8-12-1 architecture was noted to be optimal to model a 30,691 $km^2$ section of the
Ogallala Aquifer underlying the High Plains Underground Water District (HPWD) in the southern High Plains of Texas. Input
variables included (location parameters to capture hydrogeological variability), lagged water level at the well (a measure of
persistence) and water levels at four nearest neighboring wells to capture spatial dependence. Groundwater pumping — a
critical stressor — was modeled using the 3-month accumulation standardized precipitation evapotranspiration index (SPEI).

     An excellent calibration was obtained using data from the period of 1956–2009 at 119 wells within the study area. The
independent testing of the model at 38 wells over the same time-period indicated that the IST-ANN model was able to correctly
simulate groundwater level dynamics at these independent wells over the (1956–2008) time-period indicating overfitting was
not an issue. The IST-ANN model was then used to forecast water levels at 149 wells (training and testing wells) using another
independent dataset corresponding to the (2009–2015) time period. This time-period was marked by a severe and prolonged





drought which ended with a very wet year (2015) in the study area. The hydroclimatic conditions during the forecasting period were largely outside the range of variability noted during the calibration period of 1956–2008. While data-driven models such as ANNs are largely interpolation methods, the changing climate implies that these models will mostly be used in an extrapolation mode as historical hydrometeorological conditions are unlikely to be the same as those for future forecasting time-periods. While the IST-ANN model did not extrapolate as well during the forecasting time-period, the model was able to capture general trends and provided estimates that were generally better than historical averages.

Model sensitivity analysis was carried out by looking at the connection weights. SPEI (a surrogate) used for pumping was noted to be the most sensitive parameter—indicating the importance of groundwater pumping on aquifer water level predictions. The Y-coordinate was the next most sensitive parameter as it sought to capture the heterogeneity of the aquifer parameters. In particular, the N-S variability in the groundwater levels play an important role in controlling aquifer transmissivity and therefore water level predictions. The normalized nearest neighbor(s)—an indicator of spatial dependence—was noted to be more sensitive than the lagged water level at the well—hydrogeological persistence term. Water levels in the nearest neighbor affect the local hydraulic gradients and control the rate of groundwater movement to a well. These results suggest that the developed IST-ANN exhibits a high fidelity to the hydrogeological principles and conditions within the study area.

The results of the study indicate that parsimonious neural network formulations hold promise for modeling regional-scale groundwater levels in space and time. Simulating groundwater levels in agricultural regions especially under climate extremes requires an understanding of how farmers (groundwater users) adapt to drought conditions. Farmers exhibit heterogeneous behavior and choose to increase pumping or adopt conservation measures. Therefore, climate indicators, while pragmatic, may not fully capture the groundwater production preferences of farmers. Better surrogates to capture this human dimension of groundwater production is critical if data-driven models have to be extrapolated to situations that are not reflective of historical conditions to which they are calibrated to.

*Code and data availability.* The dataset and custom scripts (Python and R) developed for this study are available at https://doi.org//10.17605/OSF.IO/5KV6R (Ghaseminejad and Uddameri, 2020).





# Appendix A: Regularization of IST-ANN model

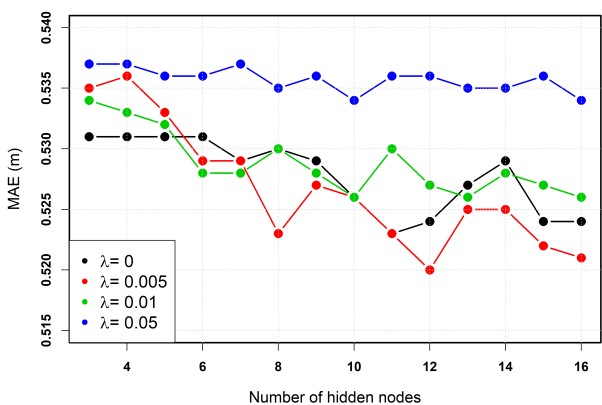

**Figure A1.** The performance of the IST-ANN model for various regularization weights and number of hidden nodes (learning rate = 0.01)

# Appendix B: Model sensitivity based on connection weights

According to Olden et al. (2004), in a model with M number of inputs (i), one output (k), and one hidden layer containing N nodes (j), The relative importance (RE) corresponding to each input can be computed as:

$$RE_i = \frac{|\sum_{j=1}^{N} w_{ij} \times w_{jk}|}{\sum_{i=1}^{M} (|\sum_{j=1}^{N} w_{ij} \times w_{jk}|)} \tag{B1}$$

Where $w_{ij}$ and $w_{jk}$ are input-hidden weights and hidden-output weights, respectively.

*Author contributions.* Conceptualization VU, AG; Formal Analysis AG; Methodology AG, VU; Software AG; Visualization AG; Validation VU; Funding Acquisition VU; Project Administration VU; Writing – Original Draft VU; Writing – Review and Editing AG, VU.

*Competing interests.* The authors declare that they have no known competing financial interests or personal relationships that could have
appeared to influence the work reported in this paper.

*Acknowledgements.* This article is based upon work that is supported by the National Institute of Food and Agriculture, U.S. Department of Agriculture, under award number 2016-68007-25066, "Sustaining Agriculture through Adaptive Management to Preserve the Ogallala aquifer under a Changing Climate".



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
