# Peer review of "Physics-inspired integrated space-time Artificial Neural Networks for regional groundwater flow modeling"

_Hydrology and Earth System Sciences, 2020_

## Referee Comment (RC1) · Anonymous Referee #1 · 7 May 2020

The manuscript presents the development of an integrated space-time Artificial Neural Network (ANN) model guided by the governing groundwater flow equation. The developed model is used to model regional groundwater flow in a portion of the Ogallala Aquifer in the Southern High Plains of Texas as an illustrative case study. The model was able to capture the general trends and provided groundwater level estimates that were better than using historical means. The methodology and the observations presented in the article is interesting and worth publishing The reference list is appropriate to the area of the investigation and up to date. A few interesting conclusions are stated in the paper. However, there are some minor concerns, but they are not very important. In my opinion the article may be published after a minor revision.

Specific comments:

Page 8: Figure 2: There is no discussion regarding Figure 2 (b). The citation of Figure 2(b) is presented in the captions of with Figure 5 and 7. I think adding a description of 2 (b) in the data compilation section will be helpful. Page 9: Line 231: I think there is some error with the citation of Table 1 here. Table 1 in the manuscript presents Performance error metrics. Page 10: Line 237: "mode"- I think it will be model. Page 13: Line 318: The preposition before "38 well" is missing. Page 14: Figure 5. And Figure 7: Y-Axis levels are not visible.

---

## Referee Comment (RC2) · Anonymous Referee #2 · 12 May 2020

The manuscript presents an integrated space-time approach for predicting one-year ahead groundwater head at multiple locations using artificial neural networks. The idea is certainly interesting and relevant for the scope of the Journal. The document is well written and the area of investigation is of extreme relevance, given the strong depletion characterizing the Southern High Plains since predevelopment.

I have only minors comments, that the authors can find below:

0- Title: considering the experiment 'physically inspired' might be a little bit of a stretch. Methodologically, the study is purely a data-driven modeling exercise with the introduction of a spatial component (the coordinates and of the neighboring wells) in the input

[Figure]

set.

1- In the introduction, some of the most recent applications of hybrid data-driven models including a spatial component are missing:

Among them, Varouchakis et al., 2019:

Varouchakis, E. A., Theodoridou, P. G., & Karatzas, G. P. (2019). Spatiotemporal geostatistical modeling of groundwater levels under a Bayesian framework using means of physical background. Journal of Hydrology, 575, 487-498

This citation can be of particular relevance since it partially addresses the 'Two-stage (ANN + interpolation) models for predicting spatiotemporal variability of groundwater levels are conceptually intuitive and pragmatic. However, this approach has limited fidelity to the groundwater system it intends to model.' issue presented by the authors.

Another recent application of spatial integration in data-driven groundwater modeling in a similar case study is constituted by Amaranto et al., 2019:

Amaranto, A., Munoz‐Arriola, F., Solomatine, D. P., & Corzo, G. (2019). A spatially enhanced data‐driven multimodel to improve semiseasonal groundwater forecasts in the High Plains aquifer, USA. Water Resources Research, 55(7), 5941-5961.

Furthermore, it is worth mentioning how Mohanty et al., (2014) developed a model for the forecasting of GW level at multiple sites:

Mohanty, S., Jha, M. K., Raul, S. K., Panda, R. K., & Sudheer, K. P. (2015). Using artificial neural network approach for simultaneous forecasting of weekly groundwater levels at multiple sites. Water Resources Management, 29(15), 5521-5532.

2 - Overall, the methodology is well presented but could benefit from the integration of an additional section, or a flowchart, explaining how the different methodological steps are interconnected to each other.

3-In the case study description, one could find interesting a comparison between the

southern portion of the High Plains Aquifer (or Ogalalla) and the remaining part. In this regard, it would also be worth mentioning some hydro-meteorological aspects that are afterward used in contextualizing results (low recharge rate etc).

4- Point comments:

Line 11 'Incorporation of spatial variability was more critical then capturing groundwater level persistence'. Not very clear

Line 256: were deemed sufficient based on autocorrelation analysis. Would be nice to see some values
* * *

---

## Editor Comment (EC1) · Dimitri Solomatine (Editor) · 16 Jun 2020

The paper is of interest to the HESS audience. Machine learning techniques in groundwater modelling are becoming more and more popular, and their closer integration with the more widely spread physically-based modelling (hibridization) is an important topic, allowing for combining the attractive features of both approaches, and better acceptance of data-driven models. Comments given by referees are quite relevant, and useful. It is clear from authors' responses, that they are aware how to address them. I wish them good luck in doing this.

117, 2020.

---

## Author Comment (AC2) · 16 Jun 2020

The manuscript presents an integrated space-time approach for predicting one-year ahead groundwater head at multiple locations using artificial neural networks. The idea is certainly interesting and relevant for the scope of the Journal. The document is well written and the area of investigation is of extreme relevance, given the strong depletion characterizing the Southern High Plains since predevelopment. I have only minors' comments, that the authors can find below:

Response: Thank you for the overall positive comments on our paper. We appreciate your thoughtful review of the manuscript and have addressed all your specific comments and concerns. Specific comments:

Comment 0: Title: considering the experiment 'physically inspired' might be a little bit of a stretch. Methodologically, the study is purely a data-driven modeling exercise with the introduction of a spatial component (the coordinates and of the neighboring wells) in the input set.

Response: Thank you for your comment. While we acknowledge that ANNs are purely data-driven models (lines 15 – 20 in the original manuscript), we also contend that their development also follow the same procedures used to parameterize physically-based groundwater flow models and be guided by the governing groundwater flow equation (line 43). In the methodology section, we start with the governing groundwater flow equation as the basis for our model development and use it to discuss the parameterization of the ANN model (e.g., line 80, line 100). We recommend that such a 'physics-inspired' model development be followed as it leads to greater transparency in the development of data-driven modeling exercises to avoid being dismissed as simply 'black boxes'. As this study has more than just a methodological focus, we feel the phrase 'physics-inspired' to be relevant as it emphasizes the importance of guiding the data-driven model development not simply on statistical considerations alone but also by physical considerations governing groundwater flow.

Comment 1-1: In the introduction, some of the most recent applications of hybrid data-driven models including a spatial component are missing: Among them, Varouchakis et al., 2019: Varouchakis, E. A., Theodoridou, P. G., & Karatzas, G. P. (2019). Spatiotemporal geostatistical modeling of groundwater levels under a Bayesian framework using means of physical background. Journal of Hydrology, 575, 487-498

This citation can be of particular relevance since it partially addresses the 'Two-stage (ANN + interpolation) models for predicting spatiotemporal variability of groundwater levels are conceptually intuitive and pragmatic. However, this approach has limited fidelity to the groundwater system it intends to model.' issue presented by the authors.

Response: Thank you for pointing us to this citation. We agree that the paper partially addresses the fidelity issues of decoupling spatial and temporal aspects of groundwater flow and have included to justify the limitations of not doing so. Line 36 of the original manuscript has been modified to include the reference as follows:

"The assumption that spatial and temporal correlations can be decoupled, is not fully consistent with conditions in the field as temporal changes in neighboring wells can locally alter flow paths and affect water level forecasts at a given well (Varouchakis et al., 2019). "

Comment 1-2: Another recent application of spatial integration in data-driven groundwater modeling in a similar case study is constituted by Amaranto et al., 2019: Amaranto, A., MunozâËŸARËĞ Arriola, F., Solomatine, D. P., & Corzo, G. (2019). A spatially enhanced dataâËŸARËĞ driven multimodel to improve semiseasonal groundwater forecasts in the High Plains aquifer, USA. Water Resources Research, 55(7), 5941-5961.

Response: Thank you for pointing to this reference. We have included this reference to further justify the use of neighboring wells while building ANN models. The following additional sentence has been added following Lines 80 – 81. "A recent study also corroborates that inclusion of information from neighboring wells improved the predictive abilities of ANN models while forecasting groundwater levels (Amaranto et al., 2019). "

Comment 1-3: Furthermore, it is worth mentioning how Mohanty et al., (2014) developed a model for the forecasting of GW level at multiple sites: Mohanty, S., Jha, M. K., Raul, S. K., Panda, R. K., & Sudheer, K. P. (2015). Using artificial neural network approach for simultaneous forecasting of weekly groundwater levels at multiple sites. Water Resources Management, 29(15), 5521-5532.

Response: Thank you for your suggestion. We have mentioned this paper when we discuss the applications of ANNs (line 20 – 21). The modified statement reads as follows: "As such, ANNs have been widely used to model groundwater level changes

at individual wells (Liu et al., 2018; Nizar Shamsuddin et al., 2017; Trichakis et al., 2011; Uddameri, 2007; Nayak et al., 2006) and simultaneously at a group of wells (Mohanty et al., 2015).

Comment 2: Overall, the methodology is well presented but could benefit from the integration of an additional section, or a flowchart, explaining how the different methodological steps are interconnected to each other.

Response: Thank for your suggestion. A workflow diagram of the research procedure has been added to the appendix (Please see attached Figure 2) and referred at line 252 of the main text and will be added to Appendix A as Figure A2 in the manuscript.

Comment 3: In the case study description, one could find interesting a comparison between the southern portion of the High Plains Aquifer (or Ogallala) and the remaining part. In this regard, it would also be worth mentioning some hydro-meteorological aspects that are afterward used in contextualizing results (low recharge rate etc.).

Response: Thank you for your comment. We have added some additional information on hydroclimatic variables as well the presence of climatic gradients at line 175 to provide context to the future discussion. We discuss limited recharge on line 200 of the manuscript. The revised paragraphs are as follows:

"Annual precipitation exhibits high temporal variability and decreases moving westward. The average maximum daily temperature varies between 21.6 C to 24.4 C with cooler temperatures noticed in the northern portions of the study area. However, summer temperatures over 37 oC can be experienced over the entire study area. Seasonal precipitation is often insufficient to meet crop water demands which are exacerbated by high temperatures during critical growth phases. Groundwater from the underlying Ogallala Aquifer is heavily used for irrigation."

Comment 4: Point comments:

• Line 11 'Incorporation of spatial variability was more critical then capturing groundwater level persistence'. Not very clear

Response: We thank the reviewer for this comment. We have modified this sentence to be clearer. The state after modification is as below: "Incorporation of spatial variability was more critical than capturing temporal persistence."

• Line 226: were deemed sufficient based on autocorrelation analysis. Would be nice to see some values

Response: Thank for your suggestion. We have added a statement that presents the numeric values of lag-4 and higher and also present a representative example in the appendix (Figure A1 in appendix A, see below). "Higher order lags were noted to have very low ($< 0.2 \pm 0.25$) autocorrelation (see Fig. A1 to be added to appendix A of the manuscript) for a representative example)."

―――――――――――――――

**Autocorrelation plot (Well C)**

[Figure: Autocorrelation plot with ACF on y-axis (0.0 to 1.0) and Lag on x-axis (values up to ~17)]

**Fig. 1.** Figure A1: Illustrative Autocorrelation Function Plot Indicating the Insignificance of Higher-Order Lags

[Figure]

**Fig. 2.** Figure A2: Workflow of the IST-ANN Modeling Procedure

---

## Author Response (AR1)

**Comment:** The manuscript presents the development of an integrated space-time Artificial Neural Network (ANN) model guided by the governing groundwater flow equation. The developed model is used to model regional groundwater flow in a portion of the Ogallala Aquifer in the Southern High Plains of Texas as an illustrative case study. The model was able to capture the general trends and provided groundwater level estimates that were better than using historical means. The methodology and the observations presented in the article is interesting and worth publishing the reference list is appropriate to the area of the investigation and up to date. A few interesting conclusions are stated in the paper. However, there are some minor concerns, but they are not very important. In my opinion the article may be published after a minor revision.

**Response:** We thank the reviewer for the overall positive comments concerning the work. We also appreciate the reviewer's specific comments and concerns. We agree with all of them and have necessary changes as stated below.

Specific comments:

**Comment:** Page 8: Figure 2: There is no discussion regarding Figure 2 (b). The citation of Figure 2(b) is presented in the captions of with Figure 5 and 7. I think adding a description of 2 (b) in the data compilation section will be helpful.

**Response:** We thank the reviewer for this comment. We agree a short description following the figure would make the presentation clear. We have added a sentence in the Data Compilation Section that explains the Wells shown in Figure 2b. The state added after line 189 is as follows:

"Twelve wells shown in Fig. 2b were selected to illustrate the temporal variability within the study area." (line 197, page 7, in revised version)

**Comment:** Page 9: Line 231: I think there is some error with the citation of Table 1 here. Table 1 in the manuscript presents Performance error metrics.

**Response:** Thank you for your comment. The line should refer to Fig 2a rather than Table 1 and we have corrected this error. (line 241, page 9, in revised version)

**Comment:** Page 10: Line 237: "mode"- I think it will be model.

**Response:** The word "mode" was used to refer to the one-step ahead forecasting approach. We have replaced the word mode with 'manner' to avoid any confusion. (line 247, page 9, in revised version)

**Comment:** Page 13: Line 318: The preposition before "38 well" is missing.

**Response:** Thank you for your detailed review. We have added the word 'at' to correct this error. (line 329, page 14, in revised version)

**Comment:** Page 14: Figure 5. And Figure 7: Y-Axis levels are not visible.

**Response:** Thank for your suggestion. We have increased the font of the axis labels to make them more visible. (Please see attached Figures). (page14 and page 16, in revised version)

[Figure]

Figure 5. Well hydrographs at select training wells during the calibration period 1956–2008 (also see Fig. 2b for well locations)

[Figure]

Figure 7. Well hydrographs for representative testing Wells within the study area (see Fig. 2b for locations of these wells; The right-hand side of the model denotes forecasting period)

**Interactive comment on "Physics-inspired integrated space-time Artificial Neural Networks for regional groundwater flow modeling"**

**by Ali Ghaseminejad and Venkatesh Uddameri**

**Anonymous Referee #2**

The manuscript presents an integrated space-time approach for predicting one-year ahead groundwater head at multiple locations using artificial neural networks. The idea is certainly interesting and relevant for the scope of the Journal. The document is well written and the area of investigation is of extreme relevance, given the strong depletion characterizing the Southern High Plains since predevelopment. I have only minors' comments, that the authors can find below:

**Response:** Thank you for the overall positive comments on our paper. We appreciate your thoughtful review of the manuscript and have addressed all your specific comments and concerns.

**Specific comments:**

**Comment 0:** Title: considering the experiment 'physically inspired' might be a little bit of a stretch. Methodologically, the study is purely a data-driven modeling exercise with the introduction of a spatial component (the coordinates and of the neighboring wells) in the input set.

**Response:** Thank you for your comment. While we acknowledge that ANNs are purely data-driven models (lines 15 – 20 in the manuscript), we also contend that their development also follow the same procedures used to parameterize physically-based groundwater flow models and be guided by the governing groundwater flow equation (line 43). In the methodology section, we start with the governing groundwater flow equation as the basis for our model development and use it to discuss the parameterization of the ANN model (e.g., line 80, line 100). We recommend that such a 'physics-inspired' model development be followed as it leads to greater transparency in the development of data-driven modeling exercises which are often dismissed as 'black boxes'.

As this study has more than just a methodological focus, we feel the phrase 'physics-inspired' to be relevant as it emphasizes the importance of guiding the data-driven model development not simply on statistical considerations but also by physical considerations governing groundwater flow. (line 44, page 2, in revised version)

**Comment 1-1:** In the introduction, some of the most recent applications of hybrid data-driven models including a spatial component are missing: Among them, Varouchakis et al., 2019:

Varouchakis, E. A., Theodoridou, P. G., & Karatzas, G. P. (2019). Spatiotemporal geostatistical modeling of groundwater levels under a Bayesian framework using means of physical background. Journal of Hydrology, 575, 487-498

This citation can be of particular relevance since it partially addresses the 'Two-stage (ANN + interpolation) models for predicting spatiotemporal variability of groundwater levels are conceptually intuitive and pragmatic. However, this approach has limited fidelity to the groundwater system it intends to model.' issue presented by the authors.

**Response:** Thank you for pointing us to this citation. We agree that the paper partially addresses the fidelity issues of decoupling spatial and temporal aspects of groundwater flow and have included to justify the limitations of doing so. Line 36 has been modified to include the reference as follows:

"The assumption that spatial and temporal correlations can be decoupled, is not fully consistent with conditions in the field as temporal changes in neighboring wells can locally alter flow paths and affect water level forecasts at a given well (Varouchakis et al., 2019). " (line 35, page 2, in revised version)

**Comment 1-2:** Another recent application of spatial integration in data-driven groundwater modeling in a similar case study is constituted by Amaranto et al., 2019:

Amaranto, A., Munozâ˘ARˇ Arriola, F., Solomatine, D. P., & Corzo, G. (2019). A spatially enhanced dataâ˘ARˇ driven multimodel to improve semiseasonal groundwater forecasts in the High Plains aquifer, USA. Water Resources Research, 55(7), 5941-5961.

**Response:** Thank you for pointing to this reference. We have included this reference to further justify the use of neighboring wells while building ANN models. The following additional sentence has been added following Lines 80 – 81.

"A recent study also corroborates that inclusion of information from neighboring wells improved the predictive abilities of ANN models while forecasting groundwater levels (Amaranto et al., 2019). " (line 86, page 3, in revised version)

**Comment 1-3:** Furthermore, it is worth mentioning how Mohanty et al., (2014) developed a model for the forecasting of GW level at multiple sites:

Mohanty, S., Jha, M. K., Raul, S. K., Panda, R. K., & Sudheer, K. P. (2015). Using artificial neural network approach for simultaneous forecasting of weekly groundwater levels at multiple sites. Water Resources Management, 29(15), 5521-5532.

**Response:** Thank you for your suggestion. We have mentioned this paper when we discuss the applications of ANNs (line 20 – 21). The modified statement reads as follows:

"As such, ANNs have been widely used to model groundwater level changes at individual wells (Liu et al., 2018; Nizar Shamsuddin et al., 2017; Trichakis et al., 2011; Uddameri, 2007; Nayak et al., 2006) and simultaneously at a group of wells (Mohanty et al., 2015). (line 22, page 1, in revised version)

**Comment 2:** Overall, the methodology is well presented but could benefit from the integration of an additional section, or a flowchart, explaining how the different methodological steps are interconnected to each other.

**Response:** Thank for your suggestion. A workflow diagram of the research procedure has been added to the appendix (Please see below) and referred at line 252. (appendix A2, page 24, in revised version)

[Figure]

Figure A2: Workflow for the IST-ANN Modeling

**Comment 3:** In the case study description, one could find interesting a comparison between the southern portion of the High Plains Aquifer (or Ogallala) and the remaining part. In this regard, it would also be worth mentioning some hydro-meteorological aspects that are afterward used in contextualizing results (low recharge rate etc.).

**Response:** Thank you for your comment. We have added some additional information on hydroclimatic variables as well the presence of climatic gradients at line 175 to provide context to the future discussion. We discuss limited recharge on line 200. The revised paragraphs are as follows:

"Annual precipitation exhibits high temporal variability and decreases moving westward. The average maximum daily temperature varies between 21.6 °C to 24.4 °C with cooler temperatures noticed in the northern portions of the study area. However, summer temperatures over 37 °C can be experienced over the entire study area. Seasonal precipitation is often insufficient to meet crop water demands which are exacerbated by high temperatures during critical growth phases.

Groundwater from the underlying Ogallala Aquifer is heavily used for irrigation." (line 178, page 7, in revised version)

**Comment 4:** Point comments:

- Line 11 'Incorporation of spatial variability was more critical then capturing groundwater level persistence'. Not very clear

**Response:** We thank the reviewer for this comment. We have modified this sentence to be clearer. The state after modification is as below:

"Incorporation of spatial variability was more critical than capturing temporal persistence." (line 11, page 1, in revised version)

- Line 226: were deemed sufficient based on autocorrelation analysis. Would be nice to see some values

**Response:** Thank for your suggestion. We have added a statement that presents the numeric values of lag-4 and higher and also present a representative example in the appendix (Figure A1 in appendix A, see below). (appendix A1, page 23, in revised version)

[revised manuscript text omitted]